# Spontaneous Neutrophil Extracellular Traps Release Are Inflammatory Markers Associated with Hyperglycemia and Renal Failure on Diabetic Retinopathy

**DOI:** 10.3390/biomedicines11071791

**Published:** 2023-06-22

**Authors:** Fátima Sofía Magaña-Guerrero, José Eduardo Aguayo-Flores, Beatriz Buentello-Volante, Karla Zarco-Ávila, Paola Sánchez-Cisneros, Ilse Castro-Salas, Enya De la Torre-Galván, José Luis Rodríguez-Loaiza, Aida Jiménez-Corona, Yonathan Garfias

**Affiliations:** 1Cell and Tissue Biology, Research Unit, Institute of Ophthalmology Conde de Valenciana, Mexico City 06800, Mexico; fatima.magana@institutodeoftalmologia.org (F.S.M.-G.); dreduardo_aguayo@hotmail.com (J.E.A.-F.); bbuentello@institutodeoftalmologia.org (B.B.-V.); karl120413@gmail.com (K.Z.-Á.); paoladejesussanchez@gmail.com (P.S.-C.); ilsecastrosalas@gmail.com (I.C.-S.); enyangelic@gmail.com (E.D.l.T.-G.); 2Department of Retina, Institute of Ophthalmology Conde de Valenciana, Mexico City 06800, Mexico; jose.rodriguez@institutodeoftalmologia.org; 3Department of Ocular Epidemiology and Visual Health, Institute of Ophthalmology Conde de Valenciana, Mexico City 06800, Mexico; aidaajc@gmail.com; 4General Directorate of Epidemiology, Health Secretariat, Mexico City 01480, Mexico; 5Department of Biochemistry, Faculty of Medicine, Universidad Nacional Autónoma de México, Mexico City 04510, Mexico

**Keywords:** hyperglycemia, diabetic retinopathy, immunopathology, inflammation markers, neutrophil, NETs

## Abstract

Diabetic retinopathy (DR) is the major microvascular complication of diabetes and causes vitreous traction and intraretinal hemorrhages leading to retinal detachment and total blindness. The evolution of diabetes is related to exacerbating inflammation caused by hyperglycemia and activation of inflammatory cells. Neutrophils are cells able to release structures of extracellular DNA and proteolytic enzymes called extracellular traps (NETs), which are associated with the persistence of inflammation in chronic pathologies. The purpose of the study was to determine the usefulness of neutrophil traps as indicators of DR progression in patients with type 2 diabetes (T2DM). We performed a case–control study of seventy-four cases classified into five groups (non-proliferative DR, mild, moderate, severe, and proliferative) and fifteen healthy controls. We found correlations between NETs and a diagnostic time of T2DM (r = 0.42; *p* < 0.0001), fasting glucose (r = 0.29; *p* < 0.01), glycated hemoglobin (HbA1c) (r = 0.31; *p* < 0.01), estimated glomerular filtration rate (eGFR) (r = −0.29; *p* < 0.01), and plasma osmolarity (r = 0.25; *p* < 0.01). These results suggest that due to NETs being associated with clinical indicators, such as HbA1c and eGFR, and that NETs are also associated with DR, clinical indicators might be explained in part through an NET-mediated inflammation process.

## 1. Introduction

Diabetic retinopathy (DR) is the main microvascular complication of diabetes and is one of the principal causes of blindness worldwide, affecting more than 93 million adults [1]. The changes that occur in the retinal tissue of patients with DR result from the development of five fundamental processes: (1) the formation of microaneurysms, (2) increased vascular permeability, (3) vascular occlusion, (4) the proliferation of neovessels and fibrous tissue in the retina and optic disc, and (5) the contraction of the vitreous and fibrovascular tissue [2].

The first ophthalmologically visible lesions appear during mild non-proliferative diabetic retinopathy (NPDR) involve the formation of capillary microaneurysms and continue with the presence or absence of diabetic macular edema (DME). DR progresses to being proliferative by the occlusion of terminal arterioles adjacent to areas with accumulations of aneurysms; at this stage, intraretinal hemorrhages and the formation of ischemic zones may occur. The secretion of proangiogenic factors and inflammatory cytokines in the retina promotes the growth of neovessels. Finally, due to the inflammatory process, contraction of the vitreous and adjacent fibrovascular tissue can lead to retinal detachment and blindness [3,4]. 

The development of DR is associated with permanent chronic inflammation caused by hyperglycemia and oxidative stress [5]. Neutrophils are inflammatory cells that are activated in the presence of microbial compounds, cytokines, and chemokines, among other inflammatory stimuli [6,7]. Activated neutrophils are defined as cells with a high expression of CD11b and CD66b and are able to perform inflammatory mechanisms such as degranulation, chemiotactic migration, and the release of neutrophil extracellular traps (NETs), among others [8,9]. NETs are structures formed by extracellular DNA decorated with proteolytic enzymes and modified proteins that are associated with the exacerbation of tissue damage in many pathologies such as sepsis [10], thrombosis [11], and autoimmune diseases such as systemic erythematosus lupus, rheumatoid arthritis, and psoriasis [12,13,14]. Thus, NETs are involved in inflammatory processes such as chronic obstructive pulmonary disease (COPD), cardiovascular risk, or risk of mortality in cancer [15,16,17].

In diabetic patients, hyperglycemia promotes NET release and high IL-6 serum levels, which are associated with nephropathy and cardiovascular complications [18]; similarly, diabetic patients have increased serum concentrations of NET compounds such as DNA histone complexes, elastase, and nucleosomes [19], suggesting that NETs and their molecular compounds could be biomarkers of diabetes [20]. The relationship between NETs and DR is not clearly understood, although some studies have reported an increased presence of circulating DNA histone and double DNA elastase complexes in DR patients compared to non-DR diabetic subjects [21,22].

The objective of this study was to evaluate the expression of neutrophil activation markers CD11b and CD66b, together with the spontaneous NET release on T2DM patients in different stages of DR to analyze the association of these inflammatory markers with hyperglycemia and renal failure and to elucidate whether these inflammatory markers are associated with DR development. 

## 2. Materials and Methods

### 2.1. Reagents

Phosphate Buffer Solution (PBS, pH 7.2), Trypan blue, p-formaldehyde, poly-L-lysine, Triton 100X, Tween 20, Propidium iodide (PI), and Hank’s Balanced Salt Solution (HBSS) were purchased from Sigma-Aldrich (Saint Louis, MI, USA). Bovine Serum Albumin (BSA) was obtained from Calbiochem (San Diego, CA, USA). Polymorphoprep was purchased from Alere Technologies AS (Jena, Germany). Purified antibody: human anti-neutrophil elastase antibody (NE) was purchased from Abcam (Cambridge, UK). FITC-conjugated anti-CD15, PE-conjugated anti-CD11b, BV450-conjugated anti-CD66b, and BD FACS Lysing 10X solutions were obtained from e-Bioscience, Beckton Dickinson (BD) (San Diego, CA, USA). An AlexaFluor-488-conjugated goat anti-rabbit antibody was obtained from LifeTechnologies (Eugene, OR, USA). Cell culture 24-well plastic plates were purchased from Corning Inc. (Corning, NY, USA). 

### 2.2. Participants

This was a case–control study conducted at the Institute of Ophthalmology “Conde de Valenciana” Foundation; subjects were recruited from October 2020 to July 2022. Healthy individuals were defined as subjects without a diagnosis of T2DM or other pathologies such as autoimmune or renal diseases, hypertension, or active infections. The study was carried out by the tenets of the Helsinki Declaration and was approved by the Institutional Review Boards of Research, Ethics, and Biosecurity of the “Conde de Valenciana” Institute of Ophthalmology (CEI-2020/01/01). All patients signed an informed consent form. 

T2DM patients were defined according to the current ADA criteria. Patients from both sexes > 18 years were included. All patients that presented inflammatory-related diseases (except diabetes), such as active or chronic infections, were excluded, as well as immunocompromised subjects. Patients with media opacity or cataracts were also excluded. Patients in which biological samples were unsuitable to work with were eliminated from the study. The selection of the patients was performed sequentially. Healthy non-diabetic patients were included as the control subjects. 

All patients underwent mydriatic fundus photography of both eyes; the images were acquired with the Optos P200DTx. Two retina specialists classified all patients according to the Diabetic Retinopathy Disease Severity Scale and the International Clinical Diabetic Retinopathy Disease Severity Scale in a blinded manner. T2DM participants were categorized into five groups: without DR; with mild non-proliferative diabetic retinopathy (NPMiDR); with moderate non-proliferative diabetic retinopathy (NPMDR); with severe non-proliferative diabetic retinopathy (NPSDR); and with proliferative diabetic retinopathy (DM-2 PDR).

Each participant donated 30 mL of peripheral blood. Urinalysis and hematic biometry were performed on each participant. Hemoglobin concentration, hematocrit, red blood cells, mean corpuscular volume, mean corpuscular hemoglobin, platelets and white blood cells (monocytes, lymphocytes, neutrophils, eosinophils and basophils), and a blood chemistry test that included glycated hemoglobin A1c (Hba1c), glucose, blood urea nitrogen (BUN), urea, uric acid, total protein, albumin, globulin, creatinine, cholesterol, triglycerides, alkaline phosphatase, electrolyte content, alanine aminotransferase (ALT), gamma-glutamyl transferase (GGT), high-density lipoproteins (HDLs), and low-density lipoproteins (LDLs) were tested.

The estimated Glomerular Filtration Rate (eGFR) was calculated using the Modification of Diet in Renal Disease (MDRD-6) formula. Plasma osmolarity refers to the serum concentration of osmotically active molecules, such as sodium (Na), potassium (K), glucose, and BUN. We use the following formula, Osm = 2(Na + K) + (Glucose/18) + (BUN/2.8), to determine plasma osmolarity.

### 2.3. Flow Cytometry Assays

Complete peripheral blood was obtained from median cubital venipuncture and collected in sodium citrate anticoagulant tubes. Immediately, 0.1 mL of blood was transferred to polystyrene flow cytometer tubes, and 15 μL of the mix of antibodies FITC-CD15 (5 μL), PE-CD11b (5 μL) and BV450-CD66b (5 μL) were added; the tubes were incubated for 30 min at 4 °C in darkness. Afterward, 0.9 mL 1X of lysing and fixing solutions (BD FACS Lysing) were added to samples and incubated for 15 min at room temperature (RT) in darkness. Finally, 104 cells of the polymorphonuclear region were acquired on a BD FACS lyric flow cytometer (BD, San Diego, CA, USA); the acquired data were analyzed using FlowJo 10.0 v software (FlowJo LLC, Ashland, OR, USA).

### 2.4. Peripheral Blood Polymorphonuclear (PMN) Cell Isolation

Human PMNs were isolated using a density gradient method. Briefly, a sample of 20 mL of complete peripheral blood from individuals fasting for 6–8 h was collected in heparin tubes. The complete blood was placed on an equal volume of a polymorphoprep solution and centrifuged at 500× *g* for 35 min at RT. The PMN ring was collected and washed with 0.8 mL of ice-cold PBS (0.1 M). The remaining erythrocytes were lysed with a lysis solution (NH_4_Cl 152.7 mM, Na_2_EDTA 0.1 mM, NaHCO_3_ 9.0 mM at pH 7.2–7.4) and washed with 0.1 M PBS as needed. The PMNs were suspended in Hank´s Balanced Salt Solution (HBSS) and kept on ice until further use. The trypan blue exclusion method was used to evaluate PMN viability. Cell purity was assessed by means of flow cytometry, obtaining ≥98% of CD11b and CD15+ double-positive cells.

### 2.5. Microscopy Staining of In Vitro NETs

The isolated PMNs from each subject were placed on poly-L-lysine pre-coated cover glasses in a 24-well plate with a density of 45 cells per well, incubated at 37 °C with an atmosphere of 5% of CO_2_ for 20 min, and let to adhere. Subsequently, cells and the spontaneous release of NETs were fixed with a 4% p-formaldehyde solution for 10 min at RT and washed twice with 0.1 M PBS. The samples were incubated with constant shaking with a blocking solution (5%-BSA, 0.1% Triton 100X, and 0.1 M PBS) for 2 h at RT °C. 

For the visualization of in vitro NET release, the samples were incubated overnight at 4 °C with constant shaking with a rabbit anti-human anti-neutrophil elastase antibody (NE) (1:100). Negative controls were performed, leaving out the primary antibody. After washing three times with 0.5 mL of PBS 0.05% tween 20 (PBS-T), the samples were incubated for 2 h with an Alexa 488 goat anti-rabbit IgG antibody (1:800) at RT °C. The samples were rinsed twice with PBS-T, and DNA visualization was performed with 25 μg/mL of propidium iodide (PI). Finally, the specimens were sealed, and the images were acquired with an ApoTome II microscope using ZEN 3.4 (blue edition) software from Carl Zeiss (Jena, Germany). 

### 2.6. Image NET Analysis

To quantify NET area release, three random images from three independent assays were analyzed from each sample with an ApoTome II microscope using Carl Zeiss ZEN 3.4 (blue edition) software. A machine learning tool was designed for the detection and quantification of NETs, as previously described, with slight modifications [23]. Briefly, to perform a homogeneous analysis, we chose only fields with 80 nuclei-stained samples with IP. A deep learning module Intellesis Trainable Segmentation from Carl Zeiss ZEN 3.4 (blue edition) for the NET area quantification was used, excluding nuclei and background. The result was considered as the total area occupied by the NET release; data were expressed as mean ± SE. 

### 2.7. Sample Size Calculation

The sample size was calculated using the mean difference of two independent groups, according to the results previously obtained by Lee, et al. [24]. A power analysis was conducted using G*power software (V 3.1.9.7, UCLA, Los Angeles, CA, USA) that was one-tailed, (1 − β) = 0.80, and α = 0.05; determining that data from 15 patients in each group were required.

### 2.8. Statistical Analysis

Beforehand, the non-parametric distribution of the data was determined by Shapiro–Wilk and Kolmogorov–Smirnov tests. Data were analyzed with non-parametric Kruskal–Wallis tests, considering *p* < 0.05 as statistically significant. Linear regressions of variables were performed with a 95% confidence interval, and one-tail non-parametric Spearman’s correlation tests were also performed. Graphics and statistical analysis were achieved using the Prism 8 GraphPad software (La Jolla, CA, USA). Finally, the association between the clinical scores with the area of NET release depending on each RD group was performed. 

## 3. Results

### 3.1. Increased Activated Neutrophil Markers and Spontaneous NET Release Relationship with the Progression Time of Diabetes in Severe Stages of DR

Demographic data and clinical evaluation of the recruited subjects are shown in Appendix A. Activated neutrophils CD15+ CD11b+ CD66b+ were found in DR groups. The population of CD15+ neutrophils of subjects with PDR had a significative higher expression of CD11b with respect to the groups with NPMiDR (** *p* < 0.01), NPSDR (* *p* < 0.05), and without T2DM (* *p* < 0.05) (Figure 1A, left panel). Similarly, in the analysis of CD66b expression on the CD15+ neutrophils population, we found that subjects with PDR had a significantly higher expression of CD66b markers with respect to subjects without T2DM and patients without DR (* *p* < 0.05). The NPMDR group showed a significantly high expression (* *p* < 0.05) of CD66b with respect to the without DR group (Figure 1A, right panel). Our results indicate that CD15+ neutrophils of patients from the NPMDR and PDR groups present a state of activation and possible inflammation by the overexpression of CD11b and CD66b. 

Additionally, we analyzed the capacity of spontaneous in vitro NET release of neutrophils in each recruited group. We found an increase in NET release in groups with any grade of DR compared with T2DM without DR and without T2DM groups (Figure 1B, left panel). The quantitative analysis showed higher NET release on NPMiDR (* *p* < 0.05), NPMDR (** *p* < 0.01), NPSDR (**** *p* < 0.0001), and PDR (** *p* < 0.01) with respect to the group without T2DM; we found significant differences on the increase in NET release in DR groups NPMDR (* *p* < 0.05), NPSDR (*** *p* < 0.001), and PDR (* *p* < 0.05) compared with the T2DM subjects without DR (Figure 1B, right panel). According to our results, the neutrophils of subjects with DR had an activated phenotype by the overexpression of CD11b and CD66b; similarly, the spontaneous in vitro NET release was higher in DR patients, suggesting a relationship between the activated/inflammatory profile of neutrophils and NETs in different stages of DR. 

We were interested in analyzing the relationship between the diagnosis time of diabetes mellitus and the activation phenotype of neutrophils. As expected, NPDR in moderate and severe stages and PDR presented a higher time (*** *p* < 0.001) of the disease diagnosis in comparison with T2DM without DR (Figure 1C). In this way, we performed correlation analyses between the time of diabetes mellitus diagnosis and the expression values (MFI) of CD11b and CD66b, as well as NET release. Although we did not find a significant correlation between the CD11b value expression and T2DM diagnosis time, we found a positive correlation between the time of diagnosis of T2DM (** *p* < 0.01, r = 0.1157) and the expression of CD66b; similarly, the time of diagnosis of T2DM correlated with NET release (**** *p* < 0.0001, r = 0.42) (Figure 1D). The release of CD66b and NETs could be associated as biomarkers of the chronic inflammation of DR until its severe stages.

### 3.2. NETs Correlated with the Hyperglycemic and Renal Status on Severe Stages of DR

The hyperglycemia and renal status indicators correlated with the inflammatory environment caused by the spontaneous in vitro NET release. Glucose is an external stimulus able to induce NET release; in this context, we found significant (*p* < 0.05) high levels of fasting glucose in all diabetic groups in comparison with non-diabetic subjects (Figure 2A). Likewise, as expected, significantly increased values of HbA1c were found in all diabetic groups in comparison with non-diabetic subjects (*p* < 0.05) (Figure 2B). 

We evaluated the correlation between two indicators of glycemic control, such as fasting glucose and HbA1c, with the in vitro NET release. We found a significant (** *p* < 0.01, r = 0.2860) positive correlation between fasting glucose (Figure 2A, right panel) and Hb1Ac (** *p* < 0.01, r = 0.3139) (Figure 2B, right panel) with NET release, suggesting that a scarce glycemic control is associated with the inflammatory environment promoting NET release, which might exacerbate the evolution of DR. 

It is well-known diabetic subjects are more prone to present vascular complications than healthy individuals. In this sense, NETs have been associated with systemic vascular and thrombotic complications that could affect organs, including kidneys. To evaluate the renal function, we calculated the eGFR and plasma osmolarity. Interestingly, subjects with the most advanced stages of retinopathy such as NPMDR, NPSDR, and PDR had significantly (*p* < 0.05) lower eGFR values compared with non-diabetic subjects (Figure 2C). Contrariwise, osmolarity values were significantly (*p* < 0.05) upregulated among all diabetic patients, including those with and without different DR in different stages (Figure 2D). 

To analyze whether the clinical indicators, such as renal functions, such as eGFR and plasma osmolarity, had a relationship with the inflammatory environment caused by NETs, we performed correlation analyses between these indicators. The results showed a significantly negative correlation (** *p* < 0.01, r = −0.2893) between NET area and eGFR, indicating that the higher level of NET, the lesser eGFR values (Figure 2C, right panel); in contrast, plasma osmolarity showed a significantly (** *p* < 0.001, r = 0.2489) positive correlation with NET release (Figure 2D, right panel). Although proteinuria values were analyzed as another clinical indicator of renal status, no significant changes were found between the analyzed groups, and no correlation was observed respect with to NET release. These results suggest that increased inflammatory conditions caused by NET release can be related to an increased renal dysfunction in subjects with DR.

### 3.3. Risk Prognosis of DR Development by Association of the Spontaneous NET Release

As we have shown, NETs are associated with eGFR, HbA1c, and the diagnostic time of T2DM, and are differentially released in DR subjects. The associated risk of development of DR was calculated considering these clinical variables and the grade of NETs. A risk matrix was designed to categorize the three clinical risk variables: the diagnostic time of T2DM, the percentage of HbA1c, and the calculated eGFR, and were assigned a score at these categories (Figure 3). According to the risk matrix, a total score was obtained from these three variables, and four levels of risk were established as follows: low (0–1), mild–moderate (2–3), moderate–increased (4), and high (5–6) (Figure 3). These data indicate that the higher risk of development of DR is related to poor glycemic control, levels of HbA1c > 9%, and eGFR values < 30 mL/min, in addition to a long time of T2DM > 9 years of diagnosis. On the other hand, the lower risk of DR development was associated with acceptable glycemic control, levels of HbA1c < 7%, eGFR values > 60 mL/min, and a DM-2 evolution of less than five years (Figure 3). 

Data in the four risk categories of clinical variables were associated with in vitro NET release to know the impact on DR development. Interestingly, the low risk of developing DR remains related to a low score together with an NET release in a range between 400 and 600. In this category, we found subjects from the groups without T2DM and T2DM without DR. On the other hand, the high risk of developing DR is present when an increased risk score is maintained along with any degree of NET release. The risk of DR development toward a moderate state of NPDR is influenced by a grade of NET release ranging between 601 and 1500; mainly, the groups associated with these risks were the subjects with DRNPMi, DRNPM, and DRNPS (Figure 3). The results altogether allow us to suggest that in vitro NET release could be considered an indicator of the risk to development DR together with other clinical variables, such as diagnostic time DM-2, HbA1c, and eGFR values.

## 4. Discussion

Neutrophils are granulocytic cells that are activated by a myriad of stimuli such as cytokines and chemokines and damage-associated molecular patrons (DAMPs), among others. The activation of neutrophils in diabetic subjects is associated with hyperglycemia and high serum lipid concentrations [25,26]. Although the precise mechanisms of neutrophils in DR development are not well understood, their activity and proteolytic granule compounds are associated with DR development and other clinical complications of diabetes mellitus [27], mainly by their role in vascular leakage and endothelium damage [28,29]. The expression of CD11b and CD66b activation markers on monocytes and neutrophils is increased in T2DM patients, promoting and increasing the adhesive capacity of neutrophils and monocytes to the endothelium and enhancing vascular and systemic damage [30]. In the present study, we found that diabetic patients with MNPDR-moderated and PDR display an increased expression of CD11b and CD66b in the CD15+ neutrophil population, suggesting that prolongated inflammation on severe grades of DR can be related to enhanced neutrophil activation. The overexpression of CD11b and CD66b has been associated with the diagnosis of inflammatory processes, such as sepsis [31], as well as the increase in migration of tumor-associated neutrophils in gastric and esophageal adenocarcinoma. They have been recognized in both conditions as prognostic factors in the diagnosis of gastric cancer [32,33] and testicular germ cell tumor [34]. Thus, the increase in the expression of both CD11b and CD66b activation markers in neutrophils could be associated as a prognostic factor of severe stages of DR. 

Neutrophil extracellular traps (NETs) are structures mainly formed by extracellular DNA decorated with nuclear-modified proteins and many proteins from granules of neutrophils. One of the diverse functions of NETs is to generate a degradant and oxidative microenvironment by their antimicrobial compounds; nevertheless, NETs also contribute to tissue damage in inflammatory and autoimmune disorders [16]. Interestingly, the peripheral neutrophils isolated from recruited subjects with NPDR and PRD showed a spontaneous capacity of NET release in vitro compared with diabetic subjects without DR and non-diabetic subjects. It is worth noting that although NET release is elevated in certain DR stages, NETs are not specific markers for diabetic retinopathy. Previously, age has been reported as a factor that influences NET release; neutrophils from young subjects are more prone to release NETs compared to neutrophils obtained from older subjects [35]. However, these findings are reported in stimulated neutrophils, and changes seen are due to failure in neutrophil activation mechanisms related to cell aging [36,37]. In the present study, although there are age differences between subjects, these differences do not impact the found data because we did not use an NET stimulator. We found changes in spontaneous NET release among subjects with DM-2 compared with diabetics with DR; all of them with similar ages. 

According to our results, fasting glucose and the HbA1c of DR subjects correlated with the in vitro NET release. It has been reported that hyperglycemia is one external stimulus that is able to induce NET release in diabetic DR subjects [22,38]. In this sense, higher levels of fast glucose and HbA1c were found in all diabetic subjects, suggesting that NET release in diabetic subjects could be used as an indicator of disease deterioration. 

It has been reported that chronic inflammation and progression of degenerative age-related eye diseases such as dry eye, glaucoma, age-related macular degeneration, and DR have an association with the participation of neutrophils in the progression and persistence of inflammatory conditions. In this way, reports have shown the overexpression of proinflammatory cytokines such as IL-6, IL-8, and IL1β and an increase in infiltrated neutrophils in diabetic rat retinas, as well as high concentrations of NET-related molecules, such as DNA protein complexes with myeloperoxidase (MPO) or neutrophil elastase (NE), on vitreous and the plasma of DR patients [21,22,39]. In this context, we found that spontaneous NETs released in the recruited groups of patients, as well as the increase in the activation markers CD11b and CD66b of neutrophils, were associated with the time to diagnose T2DM. This suggests that NETs and neutrophils would have a direct involvement in DR progression, maintaining an inflammatory environment. 

The clinical values of eGFR involved in plasmatic osmolarity and renal function are indicators of clinical conditions and the state of diabetic patients. Multiple studies suggest that NETs have a strong association with the development of vascular complications and renal diseases [40,41]. In the present study, we found that decreased eGFR levels were present in subjects with moderate and severe NPDR and PDR; likewise, eGFR values had a negative linear correlation with NET area release. Comparably, the plasma osmolarity of subjects with moderate and severe DR was increased and similarly presented a linear correlation with NET release, indicating that NET inflammation could also be associated with renal complications in DR subjects.

Interestingly, we found that the clinical variables, HbA1c, eGFR, and time of T2DM diagnosis, together with the degree of NET release, are factors associated with the development of DR. According to our results, other works have reported some markers from NETs as factors associated with adverse clinical outcomes in atherosclerosis [42]. Similarly, other reports have associated some NET markers, such as citrullinated histone (H3Citr) and DNA extracellular with glucose, IL-6, and HbA1c, in the prediction of a prothrombotic state and hypofibrinolysis during T2DM [20]. Data obtained in this study are from Mexicans; thus, extrapolations to other populations have to be made prudently. 

## 5. Conclusions

Neutrophil extracellular traps have an inflammatory mechanism of neutrophils that have a correlation with clinical variables such as HbA1c, eGFR, and time of diabetes diagnosis and are factors associated with the development of retinopathy. 

## Figures and Tables

**Figure 1 biomedicines-11-01791-f001:**
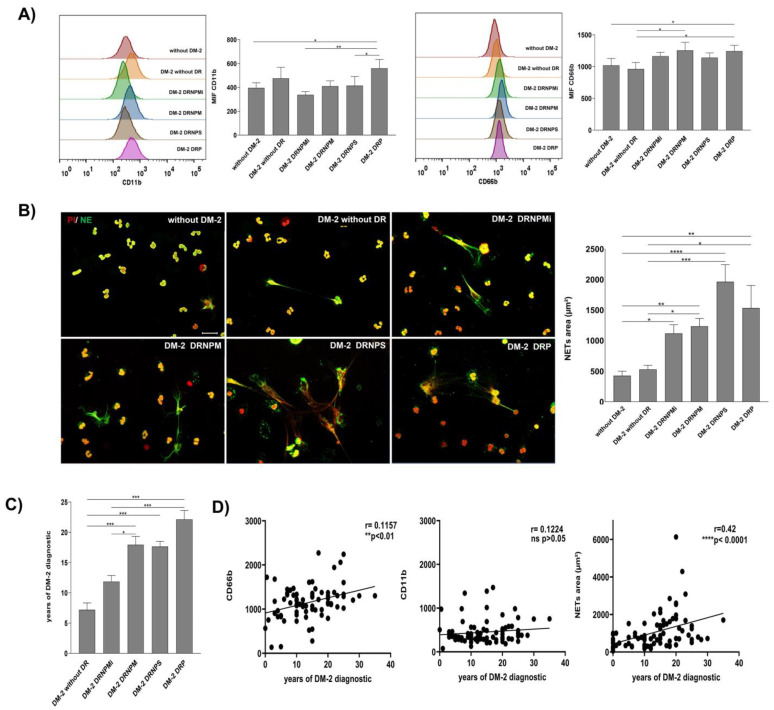
Increasingly activated neutrophil and spontaneous NET release relationship with the progression time of diabetes in severe stages of DR. Cytometry analysis of activation markers expression CD11b and CD66b on peripherical blood neutrophils (PBNs) of diabetic patients without retinopathy (DM-2), diabetic patients with a grade of non-proliferative retinopathy (mild (DM-2 NPMiDR); moderate (DM-2 NPMDR), or severe (DM-2 NPSDR), and diabetic patients with proliferative retinopathy (DM-2 PDR). A group of non-diabetic subjects (without DM-2) was recruited as a healthy healthy. Mean Fluorescence Intensity (MIF) was quantified for the expression of CD11b (left panel) and CD66b (right panel) in the PBN of diabetic patients with or without retinopathy (**A**). Micrographs of spontaneous NETs released (left panel) on diabetic patients with or without a grade of diabetic retinopathy. Neutrophil elastase (NE) staining in green was used to identify NET structures and iodide propidium in red (PI) was used to visualize extracellular DNA. The NETs are surrounded by a dashed white line. These images are representative of three random fields by the patient; scale bar = 20 μm (left panel) ((**B**), left panel). Graphical representation of the spontaneous NET area released in the groups of patients with or without retinopathy ((**B**), right panel). The spontaneous NET area was released in the groups of patients with a grade of retinopathy (DM-2 NPMiDR), (DM-2 NPMDR), (DM-2 NPSDR), and (DM-2 PDR). Data are representative from three random fields taken by the patient (*n* = 270). Graph of the progression of diabetic retinopathy dependent from the time of diagnosis of DM-2. The data are expressed as mean ± SE, * *p* < 0.05; ** *p* < 0.001; *** *p* < 0.0001; **** *p* < 0.00001, (*n* = 75) (**C**). Correlation graphs between the time of the diagnosis of DM-2 and expression of inflammatory markers CD11b and CD66b of the PBN and NETs-released area on the recruited groups. The time of the DM-2 diagnosis has a correlation with the expression of CD66b and the area of NETs released on recruited patients. The data are expressed as mean ± SE, * *p* < 0.05; ** *p*< 0.001; *** *p* < 0.0001; **** *p* < 0.00001, (*n* = 90) (**D**).

**Figure 2 biomedicines-11-01791-f002:**
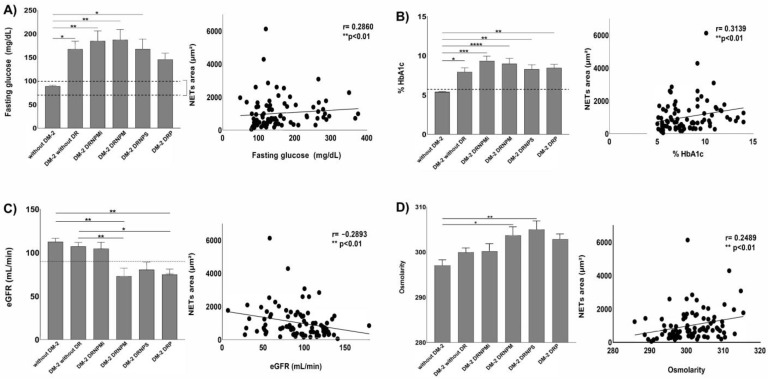
NETs are inflammatory markers correlated with hyperglycemia and renal failure in severe stages of DR. Graph of fast glucose on diabetic and retinopathy subjects; Spearman’s correlation graph between the area of NETs released and fast glucose of diabetic and retinopathy subjects (**A**). Graph of glycated hemoglobin (HbA1c) of diabetic and retinopathy subjects; Spearman’s correlation graph between the area of NETs released and HbA1c on diabetic and retinopathy subjects (**B**). Graph of estimated glomerular filtration rate (eGFR) of diabetic and retinopathy subjects; Spearman’s correlation graph between the area of NETs released and eGFR on diabetic and retinopathy subjects (**C**). Graph of plasmatic osmolarity of diabetic and retinopathy subjects; Spearman’s correlation graph between the area of NETs released and plasmatic osmolarity of diabetic and retinopathy subjects (**D**) The data are expressed as mean ± SE, * *p* < 0.05; ** *p*< 0.001; *** *p* < 0.0001; **** *p* < 0.00001 (*n* = 90). Dashed lines represent healthy normal values.

**Figure 3 biomedicines-11-01791-f003:**
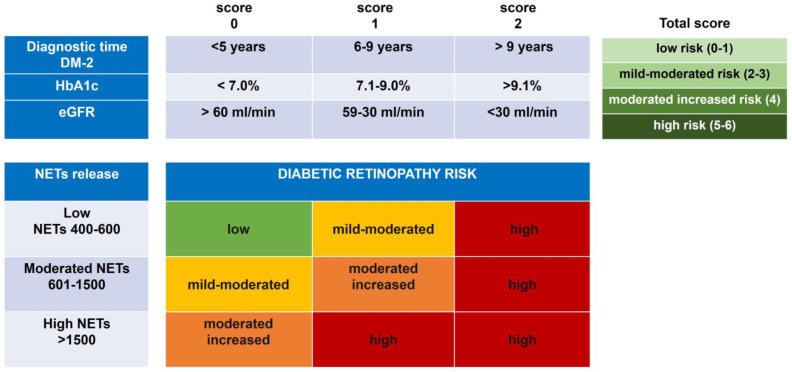
Prognostic risk of DR development by the association of the spontaneous NET release and clinical score. Matrix risk of variables associated with the development of DR. Analysis of the categorized risk factors: diagnostic time of DM-2 (<5 years, 6–9 years, and >9 years); percentage of glycated hemoglobin (HbA1c) (<7.0%, 7.1–9.0%, and >9.1%); and the calculated eGFR (>60 mL/min, 59–30 mL/min, and <30 mL/min). The total score of the diagnostic time of DM-2, the percentage of glycated hemoglobin (HbA1c), and eGFR were ranked by four risk scores (low (0–1), mild–moderate (2–3), moderate–increased (4), and high (5–6)), which were associated with the level of NET release in each group with DR.

## Data Availability

All data will be shared upon request.

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
