# Peer review of "Spontaneous Neutrophil Extracellular Traps Release Are Inflammatory Markers Associated with Hyperglycemia and Renal Failure on Diabetic Retinopathy"

_biomedicines, 2023, doi:10.3390/biomedicines11071791_

Round 1

Reviewer 1 Report

The manuscript entitled “Spontaneous neutrophil extracellular traps release are inflammatory markers associated with hyperglycemia and renal failure on diabetic retinopathy” addresses the association between neutrophil traps (structures of extracellular DNA and proteolytic enzymes; NETs) and diabetic retinopathy progression in patients with type 2 diabetes mellitus (T2DM). The authors characterized a significant association between diabetic retinopathy with NETs and HbA1c, eGFR, and diagnostic time of T2DM. The authors concluded that NETs are associated with diabetic retinopathy. The present findings are interesting, and the manuscript is clearly written.

Comments:     

1) The authors are advised to pinpoint in the discussion section that the spontaneous neutrophil extracellular traps (NETs) are not specific marker for retinopathy in DM as NETs are also increased in other pathological states. Please, discuss this limitation.

2) Caution should be applied to the conclusions of the current study since the data extracted was based on Mexican individuals only. Thus, conclusions about other ethnicities cannot be extracted. Please, discuss this point in the discussion section.

3) How did the authors decide on using the current sample size and power analysis for this study? Please, elaborate on this point and the method of calculation of sample size in the material and methods section.

4) The inclusion and exclusion criteria should be described more precisely in the current work. How were the patients selected? 

5) In section 2.5. (Microscopy staining of in vitro NETs), did the authors also perform a negative control to ensure the specific binding of the antibody to the target protein? Please, add the answer in this section.

6) In the statistical analysis section, did the authors check data normality of data (using Shapiro-Wilk test, for example) before proceeding to statistical tests? Authors are advised to address this point and add the answers in the material and methods section.

7) Given that the current study involves several groups, the Mann-Whitney test may not be appropriate for the non-parametric values as it is used to detect statistical significance between 2 groups only. The authors are advised to analyze the data using Kruskal-Wallis analysis of variance. When statistical significance is obtained, Dunn's test should be applied. The authors are advised to redo the statistical analysis for the non-parametric data as described.

8) The authors are advised to add a discussion section to the current manuscript. Please, add what is written in the conclusion section - in the current version of the manuscript - needs to be moved to the discussion section.  

9) The authors are advised to shorten the conclusion section to 1-2 paragraphs at maximum. The expected value of the conclusion section is to provide the readers with a take-home message about the study and to provide future research directions.

 Minor editing of the English language is required.

Author Response

Answers to the reviewers,

We wish to thank the reviewers for taking the time and effort necessary to provide the pertinent arguments to improve the quality of the manuscript.

All the issues asked by the reviewers where answered, giving a full explanation here, and the manuscript was modified as needed.

All modifications kindly suggested by the reviewers to the original article are now highlighted in green. We put into your consideration such changes.

Reviewer 1

The manuscript entitled “Spontaneous neutrophil extracellular traps release are inflammatory markers associated with hyperglycemia and renal failure on diabetic retinopathy” addresses the association between neutrophil traps (structures of extracellular DNA and proteolytic enzymes; NETs) and diabetic retinopathy progression in patients with type 2 diabetes mellitus (T2DM). The authors characterized a significant association between diabetic retinopathy with NETs and HbA1c, eGFR, and diagnostic time of T2DM. The authors concluded that NETs are associated with diabetic retinopathy. The present findings are interesting, and the manuscript is clearly written.

1) The authors are advised to pinpoint in the discussion section that the spontaneous neutrophil extracellular traps (NETs) are not specific marker for retinopathy in DM as NETs are also increased in other pathological states. Please, discuss this limitation.

R= We agree with this suggestion, accordingly, the following sentence has been added in the Discussion section:

“It is worth noting that although NETs release is elevated in certain DR stages, NETs are not specific markers for diabetic retinopathy.”

 2) Caution should be applied to the conclusions of the current study since the data extracted was based on Mexican individuals only. Thus, conclusions about other ethnicities cannot be extracted. Please, discuss this point in the discussion section.

R= Due to the present research is not multicenter, multinational study nor epidemiological research, the fact that the conclusions are extracted from a small size of the Mexican population implicitly indicates that extrapolations to other populations have to be made carefully. However, and to make clearer this issue, we have added the next sentence at the end of the discussion section:

“Data obtained in this research are from Mexicans, thus extrapolations to other populations have to be made prudently.”

3) How did the authors decide on using the current sample size and power analysis for this study? Please, elaborate on this point and the method of calculation of sample size in the material and methods section.

R= Thank you for this suggestion. Sample size was a priori calculated. And a new section of sample size calculation was added in materials and methods section in the new version of the manuscript.

Sample size calculation

“Sample size was calculated using mean difference of to independent groups, according to the results previously obtained by Lee, et al. [Lee KH, Cavanaugh L, Leung H, Yan F, Ahmadi Z, Chong BH, Passam F. Quantification of NETs-associated markers by flow cytometry and serum assays in patients with thrombosis and sepsis. Int J Lab Hematol. 2018 Aug;40(4):392-399. doi: 10.1111/ijlh.12800. Epub 2018 Mar 9. PMID: 29520957]. A power analysis was conducted using G*power software (UCLA, Los Angeles CA): one tailed, (1-b) = 0.80 and a=0.05; determining that data from 15 patients in each group were required.”

 4) The inclusion and exclusion criteria should be described more precisely in the current work. How were the patients selected?

R= To make this issue clear, in the Participants subsection, Materials and Methods section in the new version of the manuscript we have added the following paragraph:  

“T2DM patients were defined according to the current ADA criteria. Patients from both sexes > 18 years were included. All patients that presented inflammatory-related dis-eases (excepting diabetes), such as active or chronic infections were excluded, as well as immunocompromised subjects. Patients with media opacity or cataract were also excluded. Patients in which biological samples were unsuitable to work with, were eliminated from the study. Selection of the patients were sequentially. Healthy non-diabetic patients were included as control subjects.”

5) In section 2.5. (Microscopy staining of in vitro NETs), did the authors also perform a negative control to ensure the specific binding of the antibody to the target protein? Please, add the answer in this section.

R= Yes. To assure that the staining was specific, suitable negative controls were performed. In the new version of the manuscript, the following sentence was added.

 “Negative controls were performed leaving out primary antibody.”

6) In the statistical analysis section, did the authors check data normality of data (using Shapiro-Wilk test, for example) before proceeding to statistical tests? Authors are advised to address this point and add the answers in the material and methods section.

R= Yes. We tested a priory normality using Shapiro-Wilk and Kolmogorov-Smirnov tests, finding that most of data had non-parametric distribution.  

7) Given that the current study involves several groups, the Mann-Whitney test may not be appropriate for the non-parametric values as it is used to detect statistical significance between 2 groups only. The authors are advised to analyze the data using Kruskal-Wallis analysis of variance. When statistical significance is obtained, Dunn's test should be applied. The authors are advised to redo the statistical analysis for the non-parametric data as described.

R= We apologize for this mistake. In fact, we performed Kruskal-Wallis tests instead of Mann-Whitney tests, considering the non-parametric distribution of the majority of data and the multiple group comparisons. In the new version of the manuscript, we have amended the error. Notice that in the original version of the manuscript, we described the following: “one-tail non-parametric Spearman’s correlation tests were also performed.” Reinforcing the non-parametric distribution of the data.

8) The authors are advised to add a discussion section to the current manuscript. Please, add what is written in the conclusion section - in the current version of the manuscript - needs to be moved to the discussion section. 

R= In the new version of the manuscript we have changed numeral 5 to Discussion instead of Conclusion.

 9) The authors are advised to shorten the conclusion section to 1-2 paragraphs at maximum. The expected value of the conclusion section is to provide the readers with a take-home message about the study and to provide future research directions.

R= In the new version of the manuscript, we have added numeral 6 as a Conclusion section and a more accurate conclusion has rewritten.

Reviewer 2 Report

1- Rewrite the abstract and mention some critical numerical study results.

2- There is a typo error in this sentence: "Neutrophils are cells ablest of release structures. Please check the whole manuscript.

3- What is the importance of neutrophil-activation markers CD11b and CD66b?

4- Phosphate Buffer Solution (PBS, pH 7.2) was made in the laboratory, or was bought?

5- The conclusion is not written properly. This section must be the outcome of the results and discussions of the study. I can see some references that are showing the results and ideas of other studies. It is strongly suggested to shift that parts to the discussion and in the conclusion just express the conclusion of this study.

6- It is suggested to use the following reference to improve the manuscript :

Sabbagh, F., Muhamad, I. I., Niazmand, R., Dikshit, P. K., & Kim, B. S. (2022). Recent progress in polymeric non-invasive insulin delivery. International Journal of Biological Macromolecules.

7- some of the references are too old that need to change with some newly published papers such as refs, 2, 6, 7 ,..... Please check all the references.

Moderate editing of the English language is required.

Author Response

Answers to the reviewers,

We wish to thank the reviewers for taking the time and effort necessary to provide the pertinent arguments to improve the quality of the manuscript.

All the issues asked by the reviewers where answered, giving a full explanation here, and the manuscript was modified as needed.

All modifications kindly suggested by the reviewers to the original article are now highlighted in green. We put into your consideration such changes.

Reviewer 2.

1) Rewrite the abstract and mention some critical numerical study results.

R= We thank this suggestion and in the new version of the manuscript some statements of the abstract have been rewritten as follows.

2) There is a typo error in this sentence: "Neutrophils are cells ablest of release structures. Please check the whole manuscript.

R= We thank this suggestion; in the new version of the manuscript typo errors have been revised and corrected as needed.

3) What is the importance of neutrophil-activation markers CD11b and CD66b?

R= CD11b is the alpha-chain associated with CD18 of the beta 2- integrin Mac-1 expressed on neutrophils. This integrin participates in the regulation of many effector functions of neutrophils such as transmigration through inflammation sites, phagocytosis, and release of neutrophil extracellular traps (1). Similarly, CD66b is an adhesion molecule expressed on matured neutrophils (2) which mediates the adhesion of neutrophils to endothelial cells and is increased on inflammatory process (3). On this way the over expression of this molecules allows the activation of inflammatory effector mechanisms of neutrophils such as degranulation and release of neutrophil extracellular traps favoring inflammation and promoting endothelial damage which could promote the development of retinopathy. Both molecules are activation markers of neutrophils, which can be used as indicators of the inflammation state in our study, similarly as reported by other works and mentioned on discussion section as follows:

The overexpression of CD11b and CD66b has been associated with the diagnosis of inflammatory process such as sepsis [30], as well as the increase of migration of tumor associated neutrophils in gastric and esophageal adenocarcinoma, it has been recognized in both conditions as a prognostic factor for the diagnosis of gastric cancer [31, 32] and testicular germ cell tumor [33]. Thus, the increase on the expression of both CD11b and CD66b activation markers in neutrophils could be associated as a prognostic factor of severe stages of DR”.

4) Phosphate Buffer Solution (PBS, pH 7.2) was made in the laboratory, or was bought?

R= Phosphate Buffer Solution (PBS) was bought as a powder from Sigma-Aldrich.

5) The conclusion is not written properly. This section must be the outcome of the results and discussions of the study. I can see some references that are showing the results and ideas of other studies. It is strongly suggested to shift that parts to the discussion and in the conclusion just express the conclusion of this study.

R= We thank to the reviewer this commentary, and in the new version of manuscript a conclusion section was added as follows:

“Conclusion”

“Neutrophil extracellular traps an inflammatory mechanism of neutrophils has a correlation with the clinical variables HbA1c, eGFR and time of diabetes diagnosis and are factors associated with the development of retinopathy.”

6) It is suggested to use the following reference to improve the manuscript.

Sabbagh, F., Muhamad, I. I., Niazmand, R., Dikshit, P. K., & Kim, B. S. (2022). Recent progress in polymeric non-invasive insulin delivery. International Journal of Biological Macromolecules.

R= We appreciate your suggestion to include the aforementioned manuscript; nevertheless, after a careful revision of the interesting content, we consider not to include the article in our text because there is no a clear association among both texts.   

7-         some of the references are too old that need to change with some newly published papers such as refs, 2, 6, 7 ,..... Please check all the references.

R=We thank this commentary to the reviewer; in the new version of manuscript, we used references from not more of 10 years ago. Old references mentioned have been revised and changed as follows: 

“2. Wang, W. and A.C.Y. Lo, Diabetic Retinopathy: Pathophysiology and Treatments. Int J Mol Sci, 2018. 19(6).”

“6. Kolaczkowska, E. and P. Kubes, Neutrophil recruitment and function in health and inflammation. Nat Rev Immunol, 2013. 13(3): p. 159-75.

“7. Rada, B., Neutrophil Extracellular Traps. Methods Mol Biol, 2019. 1982: p. 517-528.”

“8. Cappenberg, A., M. Kardell, and A. Zarbock, Selectin-Mediated Signaling-Shedding Light on the Regulation of Integrin Activity in Neutrophils. Cells, 2022. 11(8).”

--

  1. Sekheri M, Othman A, Filep JG. beta2 Integrin Regulation of Neutrophil Functional Plasticity and Fate in the Resolution of Inflammation. Front Immunol. 2021;12:660760.
  2. Calzetti F, Finotti G, Tamassia N, Bianchetto-Aguilera F, Castellucci M, Cane S, et al. CD66b(-)CD64(dim)CD115(-) cells in the human bone marrow represent neutrophil-committed progenitors. Nat Immunol. 2022;23(5):679-91.
  3. Aroca R, Chamorro C, Vega A, Ventura I, Gomez E, Perez-Cano R, et al. Immunotherapy reduces allergen-mediated CD66b expression and myeloperoxidase levels on human neutrophils from allergic patients. PLoS One. 2014;9(4):e94558.